# The Normalization Method for Alleviating Pathological Sharpness in Wide Neural Networks

**Ryo Karakida**
AIST
Tokyo, Japan
karakida.ryo@aist.go.jp

**Shotaro Akaho**
AIST
Ibaraki, Japan
s.akaho@aist.go.jp

**Shun-ichi Amari**
RIKEN CBS
Saitama, Japan
amari@brain.riken.jp

## Abstract

Normalization methods play an important role in enhancing the performance of deep learning while their theoretical understandings have been limited. To theoretically elucidate the effectiveness of normalization, we quantify the geometry of the parameter space determined by the Fisher information matrix (FIM), which also corresponds to the local shape of the loss landscape under certain conditions. We analyze deep neural networks with random initialization, which is known to suffer from a pathologically sharp shape of the landscape when the network becomes sufficiently wide. We reveal that batch normalization in the last layer contributes to drastically decreasing such pathological sharpness if the width and sample number satisfy a specific condition. In contrast, it is hard for batch normalization in the middle hidden layers to alleviate pathological sharpness in many settings. We also found that layer normalization cannot alleviate pathological sharpness either. Thus, we can conclude that batch normalization in the last layer significantly contributes to decreasing the sharpness induced by the FIM.

## 1  Introduction

Deep neural networks (DNNs) have performed excellently in various practical applications [1], but there are still many heuristics and an arbitrariness in their settings and learning algorithms. To proceed further, it would be beneficial to give theoretical elucidation of how and under what conditions deep learning works well in practice.

Normalization methods are widely used to enhance the trainability and generalization ability of DNNs. In particular, batch normalization makes optimization faster with a large learning rate and achieves better generalization in experiments [2]. Recently, some studies have reported that batch normalization changes the shape of the loss function, which leads to better performance [3, 4]. Batch normalization alleviates a sharp change of the loss function and makes the loss landscape smoother [3], and prevents an explosion of the loss function and its gradient [4]. The flatness of the loss landscape and its geometric characterization have been explored in various topics such as improvement of generalization ability [5, 6], advantage of skip connections [7], and robustness against adversarial attacks [8]. Thus, it seems to be an important direction of research to investigate normalization methods from the viewpoint of the geometric characterization. Nevertheless, its theoretical elucidation has been limited to only linear networks [4] and simplified models neglecting the hierarchical structure of DNNs [3, 9].

One promising approach of analyzing normalization methods is to consider DNNs with random weights and sufficiently wide hidden layers. While theoretical analysis of DNNs often becomes intractable because of hierarchical nonlinear transformations, wide DNNs with random weights can overcome such difficulties and are attracting much attention, especially within the last few years; mean field theory of DNNs [10–14], random matrix theory [15] and kernel methods [16, 17].

They have succeeded in predicting hyperparameters with which learning algorithms work well and even used as a kernel function for the Gaussian process. In addition, recent studies on the neural tangent kernel (NTK) have revealed that the Gaussian process with the NTK of random initialization determines even the performance of trained neural networks [18, 19]. Thus, the theory of wide DNNs is becoming a foundation for comprehensive understanding of DNNs. Regarding the geometric characterization, there have been studies on the Fisher information matrix (FIM) of wide DNNs [20, 21]. The FIM widely appears in the context of deep learning [6, 22] because it determines the Riemannian geometry of the parameter space and a local shape of the loss landscape around a certain global minimum. In particular, Karakida et al. [20] have reported that the eigenvalue spectrum of the FIM is strongly distorted in wide DNNs, that is, the largest eigenvalue takes a pathologically large value (Theorem 2.2). This causes *pathological sharpness* of the landscape and such sharpness seems to be harmful from the perspective of optimization [23] and generalization [5].

In this study, we focus on the FIM of DNNs and uncover how normalization methods affect it. First, to clarify a condition to alleviate the pathologically large eigenvalues, we identify the eigenspace of the largest eigenvalues (Theorem 3.1). Then, we reveal that batch normalization in the last layer drastically decreases the size of the largest eigenvalues and successfully alleviates the pathological sharpness. This alleviation requires a certain condition on the width and sample size (Theorem 3.3), which is determined by a convergence rate of order parameters. In contrast, we find that batch normalization in the middle layers cannot alleviate pathological sharpness in many settings (Theorem 3.4) and layer normalization cannot either (Theorem 4.1). Thus, we can conclude that batch normalization in the last layer has a vital role in decreasing pathological sharpness. Our experiments suggest that such alleviation of the sharpness is helpful in making gradient descent converge even with a larger learning rate. These results give novel quantitative insight into normalization methods, wide DNNs, and geometric characterization of DNNs and is expected to be helpful in developing a further theory of deep learning.

## 2 Preliminaries

### 2.1 Model architecture

We investigate a fully-connected feedforward neural network with random weights and bias parameters. The network consists of one input layer with $M_0$ units, $L-1$ hidden layers with $M_l$ units per layer ($l = 1, 2, ..., L-1$), and one output layer:

$$u_i^l = \sum_{j=1}^{M_{l-1}} W_{ij}^l h_j^{l-1} + b_i^l, \ \ h_i^l = \phi(u_i^l), \tag{1}$$

where $h_j^0 = x_j$ are inputs. It includes a shallow network ($L = 2$) and deep ones ($L \geq 3$). We set the last layer to have a linear readout, i.e., $h_i^L = u_i^L$. The dimensionality of each variable is given by $W^l \in \mathbb{R}^{M_l \times M_{l-1}}$ and $h^l, b^l \in \mathbb{R}^{M_l}$. Suppose that the activation function $\phi(x)$ has a bounded weak derivative. A wide class of activation functions, including the sigmoid-like and (leaky-) rectified linear unit (ReLU) functions, satisfy the condition. Different layers may have different activation functions. Regarding network width, we set $M_l = \alpha_l M \ (l \leq L-1)$ and consider the limiting case of large $M$ with constant coefficients $\alpha_l$. The number of readout units is given by a constant $M_L = C$, which is independent of $M$, as is usual in practice. Suppose that the parameter set $\theta = \{W_{ij}^l, b_i^l\}$ is an ensemble generated by

$$W_{ij}^l \overset{\text{i.i.d.}}{\sim} \mathcal{N}(0, \sigma_w^2/M_{l-1}), \ \ b_i^l \overset{\text{i.i.d.}}{\sim} \mathcal{N}(0, \sigma_b^2) \tag{2}$$

then fixed, where $\mathcal{N}(0, \sigma^2)$ denotes a Gaussian distribution with zero mean and variance $\sigma^2$. We assume that there are $T$ input samples $x(t) \in \mathbb{R}^{M_0}$ ($t = 1, ..., T$) generated from an input distribution independently and that it is given by a standard normal distribution, i.e.,

$$x_j(t) \overset{\text{i.i.d.}}{\sim} \mathcal{N}(0, 1). \tag{3}$$

The FIM of a DNN is computed by the chain rule in a manner similar to that of the backpropagation algorithm:

$$\frac{\partial f_k}{\partial W_{ij}^l} = \delta_{k,i}^l h_j^{l-1}, \ \ \delta_{k,i}^l = \phi'(u_i^l) \sum_j \delta_{k,j}^{l+1} W_{ji}^{l+1}, \tag{4}$$

where we denote $f_k = u_k^L$ and $\delta_{k,i}^l := \partial f_k / \partial u_i^l$ for $k = 1, ..., C$. To avoid complicated notation, we omit index $k$ of the output unit, i.e., $\delta_i^l = \delta_{k,i}^l$.

## 2.2 Understanding DNNs through order parameters

We use the following four types of *order parameters*, i.e., $(\hat{q}_t^l, \hat{q}_{st}^l, \tilde{q}_t^l, \tilde{q}_{st}^l)$, which have been commonly used in various studies of wide DNNs [10–13, 17–20, 24]. First, we use the following order parameters for feedforward signal propagations: $\hat{q}_t^l := \sum_i h_i^l(t)^2 / M_l$ and $\hat{q}_{st}^l := \sum_i h_i^l(s) h_i^l(t) / M_l$, where $h_i^l(t)$ are the outputs of the $l$-th layer when the input is $x_j(t)$ ($t = 1, ..., T$). The variable $\hat{q}_t^l$ is the total activity of the outputs in the $l$-th layer, and the variable $\hat{q}_{st}^l$ is the overlap between the activations for different input samples $x(s)$ and $x(t)$. These variables have been utilized to explain the depth to which signals can be sufficiently propagated from the perspective of order-to-chaos phase transition [10]. In the large $M$ limit, these variables are recursively computed by integration over Gaussian distributions [10, 24]:

$$\hat{q}_t^{l+1} = \int Du \phi^2 \left( \sqrt{q_t^{l+1}} u \right), \quad \hat{q}_{st}^{l+1} = I_\phi[q_t^{l+1}, q_{st}^{l+1}], \tag{5}$$

$$q_t^{l+1} := \sigma_w^2 \hat{q}_t^l + \sigma_b^2, \quad q_{st}^{l+1} := \sigma_w^2 \hat{q}_{st}^l + \sigma_b^2, \tag{6}$$

for $l = 0, ..., L - 1$. Because input samples generated by Eq. (3) yield $\hat{q}_t^0 = 1$ and $\hat{q}_{st}^0 = 0$ for all $s$ and $t$, $\hat{q}_{st}^l$ in each layer takes the same value for all $s \neq t$, and so does $\hat{q}_t^l$ for all $t$. The notation $Du = du \exp(-u^2/2)/\sqrt{2\pi}$ means integration over the standard Gaussian density. We use a two-dimensional Gaussian integral given by $I_\phi[a, b] := \int Dy Dx \phi(\sqrt{a}x) \phi(\sqrt{a}(cx + \sqrt{1 - c^2}y))$ with $c = b/a$.

We also use the corresponding variables for backward signals: $\tilde{q}_t^l := \sum_i \delta_i^l(t)^2$ and $\tilde{q}_{st}^l := \sum_i \delta_i^l(s) \delta_i^l(t)$. The variable $\tilde{q}_t^l$ is the magnitude of the backward signals and $\tilde{q}_{st}^l$ is their overlap. Previous studies found that $\tilde{q}_{st}^l$ and $\tilde{q}_{st}^l$ in the large $M$ limit are easily computed using the following recurrence relations [11, 25],

$$\tilde{q}_t^l = \sigma_w^2 \tilde{q}_t^{l+1} \int Du \left[ \phi'(\sqrt{q_t^l} u) \right]^2, \quad \tilde{q}_{st}^l = \sigma_w^2 \tilde{q}_{st}^{l+1} I_{\phi'}[q_t^l, q_{st}^l], \tag{7}$$

for $l = 0, ..., L - 1$ with $\tilde{q}_t^L = \tilde{q}_{st}^L = 1$. Previous studies confirmed excellent agreements between these backward order parameters and experimental results [11–13]. Although these studies required the so-called *gradient independence assumption* to derive these recurrences (details are given in Assumption 3.2), Yang [25] has recently proved that such assumption is unnecessary when $\phi(x)$ has a polynomially bounded weak derivative.

The order parameters depend only on $\sigma_w^2$ and $\sigma_b^2$, the types of activation functions, and depth. The recurrence relations require $L$ iterations of one- and two-dimensional numerical integrals. They are analytically tractable in certain activation functions including the ReLUs [20].

## 2.3 Pathological sharpness of local landscapes

The FIM plays an essential role in the geometry of the parameter space and is a fundamental quantity in both statistics and machine learning. It defines a Riemannian metric of the parameter space, where the infinitesimal difference of statistical models is measured by Kullback-Leibler divergence, as in information geometry [26]. We analyze the eigenvalue statistics of the following FIM of DNNs [20, 21, 27, 28],

$$F = \sum_{k=1}^C E[\nabla_\theta f_k(t) \nabla_\theta f_k(t)^\top], \tag{8}$$

where $\theta$ is a vector composed of all parameters $\{W_{ij}^l, b_i^l\}$ and $\nabla_\theta$ is the derivative with respect to it. The average over an input distribution is denoted by $E[\cdot]$. As usual, when $T$ input samples $x(t)$ ($t = 1, ..., T$) are available for training, we replace the expectation $E[\cdot]$ with the FIM by the empirical average over $T$ samples, i.e., $E[\cdot] = \frac{1}{T} \sum_{t=1}^T$. This study investigates such an empirical FIM for arbitrary $T$. It converges to the expected FIM as $T \to \infty$. This empirical FIM is widely used in

machine learning and corresponds to the statistical model for the squared-error loss [21, 27, 28] (see Karakida et al. [20] for more details on this FIM). Recently, Kunstner et al. [29] emphasized that in the context of natural gradient algorithms, the FIM (8) leads to better optimization than an FIM approximated by using training labels.

The FIM is known to determine not only the local distortion of the parameter space but also the loss landscape around a certain global minimum. Suppose the squared loss function $E(\theta) = (1/2T) \sum_{k=1}^{C} \sum_{t=1}^{T} (y_k(t) - f_k(t))^2$, where $y_k(t)$ represents a training label corresponding to the input sample $x(t)$. The FIM is related to the Hessian of the loss function, $H := \nabla_\theta \nabla_\theta E(\theta)$, in the following manner [20, 21]:

$$H = F - \frac{1}{T} \sum_{t}^{T} \sum_{k}^{C} (y_k(t) - f_k(t)) \nabla_\theta \nabla_\theta f_k(t). \tag{9}$$

The Hessian coincides with the empirical FIM when the parameter converges to the global minimum with zero training error. In that sense, the FIM determines the local shape of the loss landscape around the minimum. This FIM is also known as the Gauss-Newton approximation of the Hessian.

Karakida et al. [20] elucidated hidden relations between the order parameters and basic statistics of the FIM's eigenvalues. We investigate DNNs satisfying the following condition.

**Definition 2.1.** *Suppose a DNN with bias terms ($\sigma_b \neq 0$) or activation functions satisfying the non-zero Gaussian mean. We refer to this as a non-centered network.*

The definition of the non-zero Gaussian mean is $\int Dz\phi(z) \neq 0$. Non-centered networks include various realistic settings because usual networks include bias terms, and widely used activation functions, such as the sigmoid function and (leaky-) ReLUs, have the non-zero Gaussian mean. Denote the FIM's eigenvalues as $\lambda_i$ ($i = 1, ..., P$) where $P$ is the number of all parameters. The eigenvalues are non-negative by definition. Their mean is $m_\lambda := \sum_i^P \lambda_i / P$ and the maximum is $\lambda_{max} := \max_i \lambda_i$. The following theorem holds:

**Theorem 2.2** ([20]). *Suppose a non-centered network and i.i.d. input samples generated by Eq. (3). When $M$ is sufficiently large, the eigenvalue statistics of $F$ are asymptotically evaluated as*

$$m_\lambda \sim \kappa_1 C/M, \;\; \lambda_{max} \sim \alpha \left( \frac{T-1}{T} \kappa_2 + \frac{\kappa_1}{T} \right) M, \tag{10}$$

*where $\alpha := \sum_{l=1}^{L-1} \alpha_l \alpha_{l-1}$, and positive constants $\kappa_1$ and $\kappa_2$ are obtained using order parameters,*

$$\kappa_1 := \sum_{l=1}^{L} \frac{\alpha_{l-1}}{\alpha} \tilde{q}_t^l \hat{q}_t^{l-1}, \;\; \kappa_2 := \sum_{l=1}^{L} \frac{\alpha_{l-1}}{\alpha} \tilde{q}_{st}^l \hat{q}_{st}^{l-1}. \tag{11}$$

The mean is asymptotically close to zero and it implies that most of the eigenvalues are very small. In contrast, $\lambda_{max}$ becomes pathologically large in proportion to the width. We refer to this $\lambda_{max}$ as *pathological sharpness* since FIM's eigenvalues determine the local shape of the parameter space and loss landscape. Empirical experiments reported that both of close-to-zero eigenvalues and pathologically large ones appear in trained networks as well [23, 30].

Pathological sharpness universally appears in various DNNs. Technically speaking, if the network is *not* non-centered (i.e., a network with no bias terms and zero-Gaussian mean; we call it a *centered network*), $\kappa_2 = 0$ holds and lower order terms of the eigenvalue statistics become non-negligible [20], and the pathological sharpness may disappear. For instance, $\lambda_{max}$ is of $O(1)$ when $T$ is properly scaled with $M$ in a centered shallow network [21]. Except for such special centered networks, we cannot avoid pathological sharpness. In practice, it would be better to alleviate the pathologically large $\lambda_{max}$ because it causes the sharp loss landscape. It requires very small learning rates (see Section 3.4) and will lead to worse generalization [4, 5]. In the following section, we reveal that a specific normalization method plays an important role in alleviating pathological sharpness.

## 3 Alleviation of pathological sharpness in batch normalization

### 3.1 Eigenspace of largest eigenvalues

Before analyzing the effects of normalization methods on the FIM, it will be helpful to characterize the cause of pathological sharpness. We find the following eigenspace of $\lambda_{max}$'s:

**Theorem 3.1.** *Suppose a non-centered network and i.i.d. input samples generated by Eq. (3). When $M$ is sufficiently large, the eigenvectors corresponding to $\lambda_{max}$'s are asymptotically equivalent to*

$$\mathrm{E}[\nabla_\theta f_k] \quad (k = 1, ..., C). \tag{12}$$

The derivation is shown in Supplementary Material A.2. This theorem gives us an idea of the effect of normalization on the FIM. Assume that we could shift the model output as $\bar{f}_k(t) = f_k(t) - \mathrm{E}[f_k]$. In this shifted model, the eigenvectors become $\mathrm{E}[\nabla_\theta \bar{f}_k] = 0$ and vanish. Naively thinking, pathologically large eigenvalues may disappear under this shift of outputs. The following analysis shows that this naive speculation is correct in a certain condition.

## 3.2 Batch normalization in last layer

In this section, we analyze batch normalization in the last layer ($L$-th layer):

$$f_k(t) := \frac{u_k^L(t) - \mu_k(\theta)}{\sigma_k(\theta)}\gamma_k + \beta_k, \tag{13}$$

$$\mu_k(\theta) := \mathrm{E}[u_k^L(t)], \quad \sigma_k(\theta) := \sqrt{\mathrm{E}[u_k^L(t)^2] - \mu_k(\theta)^2}, \tag{14}$$

for $k = 1, ..., C$. The average operator $\mathrm{E}[\cdot]$ is taken over all input samples. In practical use of batch normalization in stochastic gradient descent, the training samples are often divided into many small mini-batches, but we do not consider such division since our current interest is to evaluate the FIM averaged over all samples. We set the hyperparameter $\gamma_k = 1$ for simplicity because $\gamma_k$ only changes the scale of the FIM up to a constant. The constant $\beta_k$ works as a new bias term in the normalized network. We do not normalize middle layers ($1 \le l \le L - 1$) to observe only the effect of normalization in the last layer.

In the following analysis, we use a widely used assumption for DNNs with random weights:

**Assumption 3.2** (the gradient independence assumption [11–14, 20])**.** *When one evaluates backward order parameters, one can replace weight matrices $W^{l+1}$ in the chain rule (4) with a fresh i.i.d. copy, i.e., $\tilde{W}_{ij}^{l+1} \overset{i.i.d.}{\sim} \mathcal{N}(0, \sigma_w^2/M_l)$.*

Supposing this assumption has been a central technique of the mean field theory of DNNs [11–14, 20] to make the derivation of backward order parameters relatively easy. These studies confirmed that this assumption leads to excellent agreements with experimental results. Moreover, recent studies [25, 31] have succeeded in theoretically justifying that various statistical quantities obtained under this assumption coincide with exact solutions. Thus, Assumption 3.2 is considered to be effective as the first step of the analysis.

First, let us set $\sigma_k(\theta)$ as a constant and only consider mean subtraction in the last layer:

$$\bar{f}_k(t) := (u_k^L(t) - \mu_k(\theta))\gamma_k + \beta_k. \tag{15}$$

Since the $\sigma_k(\theta)$ controls the scale of the network output, one may suspect that the contribution of the mean subtraction would only be restrictive for alleviating sharpness. Contrary to this expectation, we find an interesting fact that the mean subtraction is essential to alleviate pathological sharpness:

**Theorem 3.3.** *Suppose a non-centered network with the mean subtraction in the last layer (Eq. (15)) and i.i.d. input samples generated by Eq. (3). In the large $M$ limit, the mean of the FIM's eigenvalues is asymptotically evaluated by*

$$m_\lambda \sim (1 - 1/T)(\kappa_1 - \kappa_2)C/M. \tag{16}$$

*The largest eigenvalue is asymptotically evaluated as follows: (i) when $T \ge 2$ and $T = O(1)$,*

$$\lambda_{max} \sim \alpha\frac{\kappa_1 - \kappa_2}{T}M, \tag{17}$$

*and (ii) when $T = O(M)$ with a constant $\rho := M/T$, under the gradient independence assumption, we have*

$$\rho\alpha(\kappa_1 - \kappa_2) + c_1 \le \lambda_{max} \le \sqrt{(C\alpha^2\rho(\kappa_1 - \kappa_2)^2 + c_2)M}, \tag{18}$$

*for non-negative constants $c_1$ and $c_2$.*

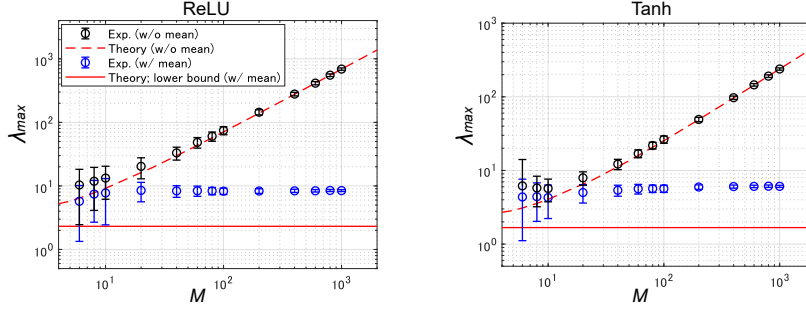

Figure 1: Effect of mean subtraction in last layer on $\lambda_{max}$. Largest eigenvalues with $T = M$ are shown. Black points show experimental values without mean subtraction, and red dashed lines show theoretical values of $\lambda_{max}$ in Theorem 2.2. In contrast, blue points show experimental values with mean subtraction and red solid lines show theoretical values of lower bound in Theorem 3.3.

The derivation is shown in Supplementary Material B.1. The mean subtraction does not change the order of $\lambda_{max}$ when $T = O(1)$. In contrast, it is interesting that it decreases the order when $T = O(M)$. The decrease in $m_\lambda$ only appears in the coefficient because $\kappa_1 > \kappa_1 - \kappa_2 > 0$ hold in the non-centered networks. Thus, we can conclude that the mean subtraction in the last layer plays an essential role in decreasing $\lambda_{max}$ when $T$ is appropriately scaled to $M$.

As shown in Fig.1, we empirically confirmed that $\lambda_{max}$ became of $O(1)$ in numerical experiments and pathological sharpness disappeared when $T = M$. Theorem 3.3 is consistent with the numerical experimental results. We numerically computed $\lambda_{max}$ in DNNs with random Gaussian weights, biases, and input samples generated by Eq. (3). We set $\alpha_l = C = 1$ and $L = 3$. Variances of parameters were given by $(\sigma_w^2, \sigma_b^2) = (2, 0)$ in the ReLU case, and $(3, 0.64)$ in the tanh case. Each points and error bars show the experimental results over 100 different ensembles. We show the value of $\rho\alpha(\kappa_1 - \kappa_2)$ as the lower bound of $\lambda_{max}$ (red line). Although this lower bound and the theoretical upper bound of order $\sqrt{M}$ are relatively loose compared to the experimental results, recall that our purpose is not to obtain the tight bounds but to show the alleviation of $\lambda_{max}$. The experimental results with the mean subtraction were much lower than those without it as our theory predicts.

From a theoretical perspective, one may be interested in how Assumption 3.2 works in the evaluation of $\lambda_{max}$. As shown in Theorem B.1 of Supplementary Material B.1, we can evaluate $\lambda_{max}$ even without using this assumption. In general, the mean subtraction in the last layer makes $\lambda_{max}$ depend on a convergence rate of backward order parameters. That is, when we have $\sum_i \delta_i^l(s)\delta_i^l(t) = \tilde{q}_{st}^l + O(1/M^q)$ with the convergence rate $q > 0$, it leads to $O(M^{1-2q^*}) \leq \lambda_{max} \leq O(M^{1-q^*})$ with $q^* = \min\{1/2, q\}$. This means that the alleviation appears for any $q$. In particular, Assumption 3.2 yields $q = q^* = 1/2$ and we obtain the lower bound of order 1. We have also confirmed that backward order parameters in numerical experiments on DNNs with random weights achieved the convergence rate of $q = q^* = 1/2$ (in Fig. S.1). Thus, Theorem 3.3 under this assumption becomes consistent with the experimental results. The batch normalization essentially requires the evaluation of the convergence rate and this is an important difference from the previous study on DNNs without normalization methods [20].

We can also add $\sigma_k^L(\theta)$ to Theorem 3.3 and obtain the eigenvalue statistics under the normalization (Eq. (13)). When $T = O(M)$, the eigenvalue statistics slightly change to

$$m_\lambda \sim Q_1(\kappa_1 - \kappa_2)/M, \quad \rho\alpha\frac{Q_2}{Q_1}(\kappa_1 - \kappa_2) + c_1' \leq \lambda_{max} \leq \sqrt{(Q_2\alpha^2\rho(\kappa_1 - \kappa_2)^2 + c_2')M} \quad (19)$$

where $Q_1 := \sum_k^C 1/\sigma_k(\theta)^2$, $Q_2 := \sum_k^C 1/\sigma_k(\theta)^4$, $c_1'$ and $c_2'$ are non-negative constants. The derivation is shown in Supplementary Material B.2. This clarifies that the variance normalization works only as a constant factor and the mean subtraction is essential to reduce pathological sharpness.

### 3.3 Batch normalization in middle layers

To distinguish the effectiveness of normalization in the last layer from those in other layers, we apply batch normalization in all layers *except for* the last layer :

$$u_i^l(t) = \sum_{j=1}^{M_{l-1}} W_{ij}^l h_j^{l-1}(t) + b_i^l, \ \ \bar{u}_i^l(t) := \frac{u_i^l(t) - \mu_i^l}{\sigma_i^l}\gamma_i^l + \beta_i^l, \ \ h_i^l(t) = \phi(\bar{u}_i^l(t)), \quad (20)$$

$$\mu_i^l := \mathrm{E}[u_i^l(t)], \ \ \sigma_i^l := \sqrt{\mathrm{E}[u_i^l(t)^2] - (\mu_i^l)^2}, \quad (21)$$

for all middle layers ($l = 1, ..., L-1$) while the last layer is kept in an un-normalized manner, i.e., $f_k(t) = \sum_j W_{kj}^L h_j^{L-1}(t) + b_k^L$. The variables $\mu_i^l$ and $\sigma_i^l$ depend on weight and bias parameters. For simplicity, we set $\gamma_i^l = 1$ and $\beta_i^l = 0$. We find a lower bound of $\lambda_{max}$ with order of $M$:

**Theorem 3.4.** *Suppose non-negative activation functions and i.i.d. input samples generated by Eq. (3). The largest eigenvalue of the FIM under the normalization (Eq. (20)) is asymptotically lower bounded by*

$$\lambda_{max} \geq \alpha_{L-1}\left(\frac{T-1}{T}\hat{q}_{st,BN}^{L-1} + \frac{\hat{q}_{t,BN}^{L-1}}{T}\right)M, \quad (22)$$

*where $\hat{q}_{t,BN}^{L-1}$ and $\hat{q}_{st,BN}^{L-1}$ are positive constants independent of $M$.*

Because the last layer is unnormalized, we can construct a lower bound composed of the activations in the $(L-1)$-th layer. Note that the set of non-negative activation functions (i.e., $\phi(x) \geq 0$) is a subclass of the non-centered networks. It includes sigmoid and ReLU functions which are widely used. The bias term, i.e., $\sigma_b^2$, does not affect the theorem because they are canceled out in the mean subtraction of each middle layer. After this batch normalization, $\lambda_{max}$ is still of $O(M)$ at lowest and the pathological sharpness is unavoidable in that sense. Thus, one can conclude that the normalization in the middle layers cannot alleviate pathological sharpness in many settings.

The constants $\hat{q}_{t,BN}^{L-1}$ and $\hat{q}_{st,BN}^{L-1}$ correspond to feedforward order parameters in batch normalization. The details are shown in Supplementary Material C.1. Although the purpose of our study was to evaluate the order of the eigenvalues, some approaches analytically compute the specific values of the order parameters under certain conditions [14, 32] (see Supplementary Material C.2 for more details). In particular, they are analytically tractable in ReLU networks as follows; $\hat{q}_{t,BN}^{L-1} = 1/2$ and $\hat{q}_{st,BN}^{L-1} = \frac{1}{2}J(-1/(T-1))$ where $J(x)$ is the arccosine kernel [14].

### 3.4 Effect on the gradient descent method

Consider the gradient descent method in a batch regime. Its update rule is given by $\theta_{t+1} \leftarrow \theta_t - \eta\nabla_\theta E(\theta_t)$ where $\eta$ is a constant learning rate. Under some natural assumptions, there exists a necessary condition of the learning rate for the gradient dynamics to converge to a global minimum [20, 23];

$$\eta < 2/\lambda_{max}. \quad (23)$$

Because our theory shows that batch normalization in the last layer decreased $\lambda_{max}$, the appropriate learning rate for convergence becomes larger. To confirm this effect on the learning rate, we did experiments on training with the gradient descent as shown in Fig. 2. we trained DNNs with various widths by using various fixed learning rates, providing i.i.d. Gaussian input samples and labels generated by corresponding teacher networks. It was the same setting as the experiment shown in [20]. Fig. 2 (left) shows the color map of training losses without any normalization method and is just a reproduction of [20]. Losses exploded in the gray area (i.e., were larger than $10^3$) and the red line shows the theoretical value of $2/\lambda_{max}$, which was calculated with the FIM at random initialization. Training above the red line exploded in sufficiently widen DNNs, just as the necessary condition (23) predicts. In contrast, Fig. 2 (right) shows the result of the batch normalization (mean subtraction) in the last layer. We confirmed that it allows larger learning rates for convergence and they are independent of width. We calculated the theoretical line by using the lower bound of $\lambda_{max}$, i.e., $\eta = 2/(\rho\alpha(\kappa_1 - \kappa_2))$. Note that Fig. 2 shows the results on the single trial of training with fixed initialization. It caused the stripe pattern of color map depending on the random seed of each width, especially in the case of normalized networks. As shown in Fig. S.2 of Supplementary Material

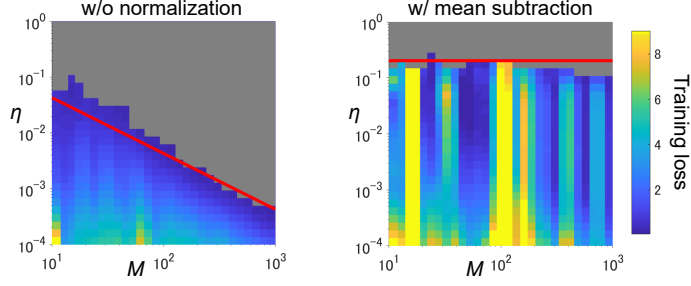

Figure 2: Exhaustively searched training losses depending on $M$ (width) and $\eta$ (learning rate). The color bar shows the value of training loss after 1000 steps of training. We trained deep ReLU networks with $\alpha_l = C = 1$, $L = 3$, $T = 1000$ and $(\sigma_w^2, \sigma_b^2) = (4, 1)$.

D, accumulation of multiple trials achieves lower losses regardless of the width. Thus, the batch normalization is helpful to set larger learning rates, which could be expected to speed-up the training of neural networks [2].

## 4    Pathological sharpness in layer normalization

It is an interesting question to investigate the effect of other normalization methods on pathological sharpness. Let us consider layer normalization [9]:

$$u_i^l(t) = \sum_{j=1}^{M_{l-1}} W_{ij}^l h_j^{l-1}(t) + b_i^l, \ \ \bar{u}_i^l(t) := \frac{u_i^l(t) - \mu^l(t)}{\sigma^l(t)}\gamma_i^l + \beta_i^l, \ \ h_i^l = \phi(\bar{u}_i^l(t)), \qquad (24)$$

$$\mu^l(t) := \sum_i u_i^l(t)/M_l, \ \ \sigma^l(t) := \sqrt{\sum_i u_i^l(t)^2/M_l - \mu^l(t)^2}, \qquad (25)$$

for all layers ($l = 1, ..., L$). The network output is normalized as $f_k(t) = \bar{u}_k^L(t)$. While batch normalization (20) normalizes the pre-activation of each unit across batch samples, layer normalization (24) normalizes that of each sample across the units in the same layer. Although layer normalization is the method typically used in recurrent neural networks, we show its effectiveness in feedforward networks to contrast the effect of batch normalization on the FIM. For simplicity, we set $\gamma_i^l = 1$ and $\beta_i^l = 0$. Then, we find

**Theorem 4.1.** *Suppose a non-centered network, i.i.d. input samples generated by Eq. (3), and the gradient independence assumption. When $M$ is sufficiently large and $C > 2$, the eigenvalue statistics of the FIM under the normalization (Eq. (24)) are asymptotically evaluated as*

$$m_\lambda \sim (C-2)\eta_1 \kappa_1'/M, \ \ \frac{\alpha s}{(C-2)\eta_1 \kappa_1'}M \le \lambda_{max} \le \alpha\sqrt{s}M, \qquad (26)$$

*with*

$$s = \frac{(\eta_3^2 - \eta_1^2)T + (C-2)(\eta_1^2 T - \eta_2)}{T}\kappa_2'^2 + (C-2)\eta_2 \frac{\kappa_1'^2}{T}, \qquad (27)$$

*where $\kappa_1'$, $\kappa_2'$ and $\eta_i$ ($i = 1, 2, 3$) are constants independent of $M$.*

Layer normalization does not alleviate the pathological sharpness in the sense that $\lambda_{max}$ is of order $M$. Intuitively, this is because $\mathrm{E}[f_k]$ is not equal to $\sum_k f_k/C$ and the mean subtraction in the last layer does not cancel out the eigenvectors in Theorem 3.1. We can compute $\kappa_1'$ and $\kappa_1'$ by using order parameters and $\eta_i$ ($i = 1, 2, 3$) by the variance $\sigma^L(t)^2$. The definition of each variable and the proof of the theorem are shown in Supplementary Material E. The independence assumption is used in derivation of backward order parameters as usual [11, 14]. When $C = 2$, the FIM becomes a zero matrix because of a special symmetry in the last layer. Therefore, the non-trivial case is $C > 2$.

## 5    Related work

**Normalization and geometric characterization.** Batch normalization is believed to perform well because it suppresses the internal covariate shift [2]. Recent extensive studies, however, have reported

alternative explanations on how batch normalization works [3, 4]. Santurkar et al. [3] empirically found that batch normalization decreases a sharp change of the loss function and makes the loss landscape smoother. Bjorck et al. [4] reported that batch normalization works to prevent an explosion of the loss and gradients. While some theoretical studies analyzed FIMs in un-normalized DNNs [6, 20, 21], analysis in normalized DNNs has been limited. Santurkar et al. [3] analyzed gradients and Hessian under batch normalization in a single layer and theoretically evaluated their worst case bounds, but its inequality was too general to quantify the decrease of sharpness. In particular, it misses the special effect of the last layer, as we found in this study. The original paper [9] of layer normalization analyzed the FIM in generalized linear models (GLMs) and argued that the normalization could decrease curvature of the parameter space. While a GLM corresponds to the single layer model, shallow and deep networks have hidden layers. As the hidden layers become wide, pathological sharpness appears and layer normalization suffers from it.

**Gradient descent method.** There are other related works in addition to those mentioned in Section 3.4. Bjorck et al. [4] speculated that larger learning rates realized by batch normalization may help stochastic gradient descent avoid sharp minima and it leads to better generalization. Wei et al. [32] estimated $\lambda_{max}$ and $\eta$ under a special type of batch-wise normalization. Because their normalization method approximates a chain rule of backpropagation by neglecting the contribution of mean subtraction, it suffers from pathological sharpness and requires smaller learning rates.

**Neural tangent kernel.** The FIM and NTK satisfy a kind of duality, and share the same non-zero eigenvalues. Our proofs on the eigenvalue statistics use NTK with standard parameterization, i.e., $F^*$ in Supplementary Material A.1. The NTK at random initialization is known to determine the gradient dynamics of a sufficiently wide DNN in function space. The sufficiently wide network can achieve a zero training error and it means that there is always a global minimum sufficiently close to random initialization. In the parameter space, Lee et al. [19] proved that NTK dynamics is sufficiently approximated by the gradient descent of a linearized model expanded around random initialization $\theta_0$: $f(x; \theta_t) = f(x; \theta_0) + \nabla_\theta f(x; \theta_0)^\top \omega_t$, where $\omega_t := \theta_t - \theta_0$ and $t$ means the step of the gradient descent. Naively speaking, this suggests that the optimization of the wide DNN approximately becomes convex and the loss landscape is dominated by a quadratic form with the FIM, i.e., $\omega_t^\top F \omega_t$.

# 6 Discussion

There remain a number of directions for extending our theoretical framework. Recent studies on wide DNNs have revealed that the NTK of random initialization dominates the training dynamics and even the performance of trained networks [18, 19]. Since the NTK is defined as a right-to-left reversed Gram matrix of the FIM under a special parameterization, the convergence speed of the training dynamics is essentially governed by the eigenvalues of the FIM at the random initialization. Analyzing these dynamics under normalization remains to be uncovered. For further analysis, random matrix theory will also be helpful in obtaining the whole eigenvalue spectrum or deriving tighter bounds of the largest eigenvalues. Although random matrix theory has been limited to a single layer or shallow networks [21], it will be an important direction to extend it to deeper and normalized networks.

There may be potential properties of normalization methods that are not detected in our framework. Kohler et al. [33] analyzed the decoupling of the weight vector to its direction and length as in batch normalization and weight normalization. They revealed that such decoupling could contribute to accelerating the optimization. Bjorck et al. [4] discussed that deep linear networks without bias terms suffer from the explosion of the feature vectors and speculated that batch normalization is helpful in reducing this explosion. This implies that batch normalization may be helpful to improve optimization performance even in a centered network. Yang et al. [14] developed an excellent mean-field framework for batch normalization through all layers and found that the gradient explosion is induced by batch normalization in networks with extreme depth. Even if batch normalization alleviates pathological sharpness regarding the width, the coefficients of order evaluation can become very large when the network is extremely deep. It may cause another type of sharpness. It is also interesting to explore SGD training under normalization and quantify how the alleviation of sharpness affects appropriate sizes of learning rate and mini-batch, which have been mainly investigated in SGD training without normalization [34]. Further studies on such phenomena in wide DNNs would be helpful for further understanding and development of normalization methods.

**Acknowledgments**

This work was partially supported by a Grant-in-Aid for Young Scientists (19K20366) from the Japan Society for the Promotion of Science (JSPS).

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
