[Supplementary Material · NIPS2019_camera_full_supp(2).pdf]

# Supplementary Materials

## A  Basic statistics of FIM without normalization

### A.1  Reversed FIM

We prepare the following two lemmas to prove the theorems in the main text.

An FIM is a $P \times P$ matrix, where $P$ is the dimension of all parameters. Define a $P \times CT$ matrix $R$ by

$$R \quad := \quad \frac{1}{\sqrt{T}}[\nabla_\theta f_1 \ \ \nabla_\theta f_2 \ \ \cdots \ \ \nabla_\theta f_C]. \tag{S.1}$$

Its columns are the gradients on each input, i.e., $\nabla_\theta f_k(t)$ ($t = 1, ..., T$). One can represent an empirical FIM by

$$F = RR^\top. \tag{S.2}$$

Let us refer to the following $CT \times CT$ matrix as a reversed FIM:

$$F^* := R^\top R, \tag{S.3}$$

which is the right-to-left reversed Gram matrix of $F$. This $F^*$ is essentially the same as the NTK [18]. The $F$ and $F^*$ have the same non-zero eigenvalues by definition. Karakida et al. [20] introduced $F^*$ to derive the eigenvalue statistics in Theorem 2.2. Technically speaking, they derived the eigenvalue statistics under the gradient independence assumption (Assumption 3.2). However, Yang [25] recently succeeded in proving that this assumption is unnecessary. Therefore, Theorem 2.2 is free from this assumption.

To evaluate the effects of batch normalization, we need to take a more careful look into $F^*$ than done in previous studies. As shown in Supplementary Material B, the FIM under batch normalization in the last layer requires information on how fast backward order parameters asymptotically converge in the large $M$ limit. Let us introduce the following variables depending on $M$:

$$\tilde{q}_{M,t}^l := \sum_i \delta_i^l(t)^2, \ \ \tilde{q}_{M,st}^l := \sum_i \delta_i^l(s)\delta_i^l(t). \tag{S.4}$$

When $C$ is a constant of order 1, $\delta_i^l(t)$ is of order $1/\sqrt{M}$ and the above summations become of order 1. The variable $\tilde{q}_{M,t}^l$ is a special case of $\tilde{q}_{M,st}^l$ with $s = t$. Recent studies [25, 31] proved that, in the large $M$ limit, backward order parameters asymptotically converge to $\tilde{q}_{st}^l$ satisfying the recurrence relations (Eq. (7)). Suppose that we have in the large $M$ limit,

$$\tilde{q}_{M,st}^l = \tilde{q}_{st}^l + O(1/M^q), \tag{S.5}$$

where $q > 0$ determines a convergence rate. Schoenholz et al. [11] derived $\tilde{q}_{\infty,st}^l = \tilde{q}_{st}^l$ under the gradient independence assumption. Yang [25] succeeded in deriving the recurrence relations without using the gradient independence assumption. It also gave an upper bound of the residual term $O(1/M^q)$ although an explicit value of $q$ was not shown. Arora et al. [31] also succeeded in deriving the recurrence relations in ReLU networks by using a non-asymptotic method and obtained $q \geq 1/4$ (Theorem B.2 in [31]). Thus, previous studies have paid much attention to $\tilde{q}_{st}^l$ while there has been almost no comprehensive discussion regarding the residual term $O(1/M^q)$. As shown in Supplementary Material B.1.3, we confirmed that $q = 1/2$ holds in simulations of typical DNN models and that the gradient independence assumption yields $q = 1/2$.

Between the reversed FIM and convergence rate $q$, we found that the following lemma holds. This lemma is a minor extension of Supplementary Material A in [20] into the case without the gradient independence assumption.

**Lemma A.1.** *Suppose a non-centered network and i.i.d. input samples generated by Eq. (3). When $M$ is sufficiently large, the $F^*$ can be partitioned into $C^2$ block matrices whose $(k, k')$-th block is a $T \times T$ matrix defined by*

$$F^*(k, k') = \alpha \frac{M}{T} K \delta_{kk'} + \frac{1}{T} O(M^{1-q^*}), \tag{S.6}$$

where $q^* = \min\{q, 1/2\}$, $k, k' = 1, ..., C$ and $\delta_{kk'}$ is the Kronecker delta. The matrix $K$ has entries given by

$$K_{st} = \kappa_1 \ (s = t), \ \kappa_2 \ (s \neq t). \tag{S.7}$$

*Proof.* We have the parameter set $\theta = \{W_{ij}^l, b_i^l\}$ but the number of bias parameters (of $O(M)$) is much less than that of weight parameters (of $O(M^2)$). Therefore, the contribution of the FIM corresponding the bias terms are negligibly small in the large $M$ limit [20], and what we should analyze is weight parts of the FIM, that is, $\nabla_{W_{ij}^l} f_k$. The $(k, k')$-th block of $F^*$ has the $(s, t)$-th entry as

$$F^*(k, k')_{st} = \sum_l \sum_{ij} \nabla_{W_{ij}^l} f_k^\top (s) \nabla_{W_{ij}^l} f_{k'}(t)/T \tag{S.8}$$

$$= \sum_l M_{l-1}(\sum_i \delta_{k,i}^l(s)\delta_{k',i}^l(t))\hat{q}_{M,st}^{l-1}/T \tag{S.9}$$

for $s, t = 1, ..., T$. In the large $M$ limit, we can apply the central limit theorem to the feedforward propagation because the pre-activation $u_i^l$ is a weighted sum of independent random weights [10, 24]: $\hat{q}_{M,st}^l = \hat{q}_{st}^l + O(1/\sqrt{M})$. This convergence rate of $1/2$ is also known in non-asymptotic evaluations [16, 31]. We then have

$$F^*(k, k')_{st} = \sum_l M_{l-1}(\tilde{q}_{st}^l\delta_{kk'} + O(1/M^q))(\hat{q}_{st}^{l-1} + O(1/\sqrt{M}))/T \tag{S.10}$$

$$= \sum_l \frac{M_{l-1}}{T} \tilde{q}_{st}^l\hat{q}_{st}^{l-1}\delta_{kk'} + \frac{1}{T}O(M^{1-q*}) \tag{S.11}$$

$$= \alpha\kappa_2\frac{M}{T}\delta_{kk'} + \frac{1}{T}O(M^{1-q*}), \tag{S.12}$$

where $q^* := \min\{q, 1/2\}$. The backward order parameters for $k \neq k'$ become zero because the chains do not share the same weight $W_{ki}^L$ and the initialization of the recurrence relations (Eq. (7)) becomes $\tilde{q}_{st}^L = \sum_i \delta_{k,i}^L(s)\delta_{k',i}^L(t) = \sum_i \delta_{ki}\delta_{k'i} = 0$. When $t = s$, we have $F^*(k, k')_{tt} = \alpha\kappa_1 M/T\delta_{kk'} + O(M^{1-q^*})/T$. □

The current work essentially differs from [20] in the point that the evaluation of $F^*$ includes the convergence rate. The previous work investigated DNNs without any normalization method and such cases allow us to focus on the first term of the right-had side of Eq. (S.6). This is because the second term becomes asymptotically negligible in the large $M$ limit. In contrast, batch normalization in the last layer makes the first term comparable to the second term and requires careful evaluation of the second term. Thus, eigenvalues statistics become dependent on the convergence rate.

The previous work [20] showed that the matrix $K$ in the first term of (S.6) determines the eigenvalue statistics such as $m_\lambda$ and $\lambda_{max}$ in the large $M$ limit. The assumption of i.i.d. input samples makes the structure of matrix $K$ easy to analyze, i.e., all the diagonal terms take the same $\kappa_1$ and all the non-diagonal terms take $\kappa_2$. Using this matrix $K$, we can also derive the eigenvectors of $F^*$ corresponding to $\lambda_{max}$:

**Lemma A.2** (Supplementary Material A.4 in [20]). *Suppose a non-centered network and i.i.d. input samples generated by Eq. (3). Denote the eigenvectors of $F^*$ corresponding to $\lambda_{max}$ as $\nu_k \in \mathbb{R}^{CT}$ $(k = 1, ..., C)$. In the large $M$ limit, they are asymptotically equivalent to*

$$(\nu_k)_i := \begin{cases} 1/\sqrt{T} & ((k-1)T + 1 \leq i \leq kT), \\ 0 & (\text{otherwise}). \end{cases} \tag{S.13}$$

It should be remarked that the above results require $\kappa_2 > 0$. Technically speaking, the second term of Eq. (S.6) is negligible because $\kappa_1$ is positive by definition and $\kappa_2$ is also positive in a non-centered network. If one considers a centered network, however, the initialization of recurrence relations, i.e., $\hat{q}_{st}^0 = 0$, recursively yields

$$q_{st}^{l+1} = \sigma_w^2 I_\phi[q_t^l, 0] = \sigma_w^2 \int DyDx\phi(\sqrt{q_t^l}x)\phi(\sqrt{q_t^l}y) = 0, \tag{S.14}$$

and it gives $\hat{q}_{st}^l = 0$ for all $l$ and $\kappa_2 = 0$. In such cases, the second term of Eq. (S.6) dominates the non-diagonal entries of $F^*(k, k')$ and affects the eigenvalue statistics. In contrast, we have $q_{st}^l > 0$ and $\hat{q}_{st}^l > 0$ in a non-centered network. Because $\tilde{q}_{st}^l > 0$ holds as well, we have $\kappa_2 > 0$ and the second term becomes negligible in the large $M$ limit.

## A.2 Eigenspace of $\lambda_{max}$

To prove Theorem 3.1, we use the eigenvector $\nu_k$ obtained in Lemma A.2. The eigenspace of $F$ corresponding to $\lambda_{max}$ is constructed from $\nu_k$. Let us denote an eigenvector of $F$ as $v$ satisfying $Fv = \lambda_{max}v$. By multiplying $R^\top$ by both sides, we have

$$F^*(R^\top v) = \lambda_{max}(R^\top v). \tag{S.15}$$

This means that $R^\top v$ is the eigenvector of $F^*$. Then, we obtain $R^\top v = \sum_k c_k \nu_k$ up to a scale factor by using coefficients $c_k$ satisfying $\sum_k c_k = 1$. Substituting it into $(RR^\top)v = \lambda_{max}v$, we have $v = \sum_k c_k R\nu_k = \sum_k c_k \mathrm{E}[\nabla_\theta f_k]$. As a result, the eigenspace of $F$ corresponding to $\lambda_{max}$ is spanned by $\mathrm{E}[\nabla_\theta f_k]$. It is easy to conform that the derivative $\nabla_{W_{ij}^l} f_k (= \delta_i^l h_j^{l-1})$ is of $O(1/\sqrt{M})$ and we have

$$F \cdot \mathrm{E}[\nabla_\theta f_k] = \lambda_{max}\mathrm{E}[\nabla_\theta f_k] + O(M^{1/2-q^*}), \tag{S.16}$$

for $k = 1, ..., C$. The first term of the right-hand side of Eq. (S.16) is of $O(M^{1/2})$ in non-centered networks and asymptotically larger than the second term. Thus, we obtain Theorem 3.1.

Note that Lemma A.2 requires non-centered networks and so does Theorem 3.1. Pathological sharpness appears because non-centered networks make the first term of Eq. (S.16) non-negligible. In contrast, centered networks have $\kappa_2 = 0$ and the order of the above $\lambda_{max}$ becomes lower for a sufficiently large $T$. In such case, the second term of Eq. (S.16) becomes dominant and we cannot judge whether $\mathrm{E}[\nabla_\theta f_k]$ is the eigenvector of FIM or not.

# B Batch normalization in last layer

## B.1 Mean subtraction

### B.1.1 Mean of eigenvalues

The FIM under the mean subtraction (Eq. (15)) is expressed by

$$F_{L,mBN} = \sum_k \mathrm{E}[\nabla_\theta \bar{f}_k(t)\nabla_\theta \bar{f}_k(t)^\top] = (R - \bar{R})(R - \bar{R})^\top, \tag{S.17}$$

where $\bar{R}$ is a $CT \times P$ matrix whose $k$-th column is given by a vector $\nabla_\theta \mu_i/\sqrt{T}$ $((i-1)T + 1 \leq k \leq iT, i = 1, 2, ..., C)$. Note that the hyperparameter $\beta_k$ disappears since $\beta_k$ is independent of $\theta$. Here, we define the projector

$$G := I_T - 1_T(1_T)^\top/T, \tag{S.18}$$

which satisfies $G^2 = G$. Using this projector, we have $RG = R - \bar{R}$ and

$$F_{L,mBN} = R(I_C \otimes G)R^\top, \tag{S.19}$$

where $I_C$ is a $C \times C$ identity matrix and $\otimes$ is the Kronecker product. We introduce a reversed Gram matrix of the FIM under the mean subtraction:

$$F_{L,mBN}^* := (R - \bar{R})^\top(R - \bar{R}) = (I_C \otimes G)F^*(I_C \otimes G)^\top. \tag{S.20}$$

Let us partition $F_{L,mBN}^*$ into $C^2$ block matrices and denote its $(k, k')$-th block as a $T \times T$ matrix $F_{L,mBN}^*(k, k')$. Substituting the $F^*$ (S.6) into the above, we obtain these blocks as

$$F_{L,mBN}^*(k, k') = \alpha\frac{M}{T}K_{L,mBN}\delta_{kk'} + \frac{1}{T}O(M^{1-q^*}), \tag{S.21}$$

with

$$(K_{L,mBN})_{st} := \begin{cases} (\kappa_1 - \kappa_2)(1 - 1/T) & (s = t), \\ -(\kappa_1 - \kappa_2)/T & (s \neq t). \end{cases} \tag{S.22}$$

We assume $T \geq 2$ since $T = 1$ is trivial.

The mean of eigenvalues is asymptotically obtained by

$$m_\lambda = \mathrm{Trace}(F_{L,mBN}^*)/P \tag{S.23}$$

$$\sim C(1 - 1/T)(\kappa_1 - \kappa_2)/M. \tag{S.24}$$

**B.1.2 Largest eigenvalue when $T \geq 2$ and $T = O(1)$**

First, we obtain a lower bound of $\lambda_{max}$. In general, we have

$$\lambda_{max} = \max_{v; ||v||^2 = 1} v^\top F_{L,mBN}^* v. \tag{S.25}$$

We then find

$$\lambda_{max} \geq \nu^\top F_{L,mBN}^* \nu, \tag{S.26}$$

where $\nu$ is a $CT$-dimensional vector whose $((i-1)T+1)$-th entries are $1/\sqrt{2C}$, $((i-1)T+2)$-th entries are $-1/\sqrt{2C}$, and the others are 0 ($i = 1, ..., C$). We then have

$$\lambda_{max} \geq \alpha \frac{\kappa_1 - \kappa_2}{T} M + O(M^{1-q^*}). \tag{S.27}$$

Next, we obtain an upper bound of $\lambda_{max}$. In general, the maximum eigenvalue is denoted as the spectral norm $|| \cdot ||_2$, i.e., $\lambda_{max} = ||F_{L,mBN}^*||_2$. Using the triangle inequality, we have

$$\lambda_{max} \leq ||\bar{F}_{L,mBN}^*||_2 + ||\tilde{F}_{L,mBN}^*||_2. \tag{S.28}$$

We divided $F_{L,mBN}^*$ into $\bar{F}_{L,mBN}^*$ corresponding to the first term of Eq. (S.21) and $\tilde{F}_{L,mBN}^*$ corresponding to the second term. $\bar{F}_{L,mBN}^*$ is composed of $\alpha \frac{M}{T} K_{L,mBN}$. The eigenvalues of $\alpha \frac{M}{T} K_{L,mBN}$ are explicitly obtained as follows: $\lambda_1 = 0$ for an eigenvector $e = (1, ..., 1)$, and $\lambda_i = \alpha(\kappa_1 - \kappa_2)M/T$ for eigenvectors $e_1 - e_i$ ($i = 2, ..., T$), where $e_i$ denotes a unit vector whose entries are 1 for the $i$-th entry and 0 otherwise. We then obtain

$$||\bar{F}_{L,mBN}^*||_2 = \alpha(\kappa_1 - \kappa_2)M/T. \tag{S.29}$$

Each entry of $\tilde{F}_{L,mBN}^*$ is of $O(M^{1-q^*})$. We then have

$$||\tilde{F}_{L,mBN}^*||_2 \leq ||\tilde{F}_{L,mBN}^*||_F = O(M^{1-q^*}), \tag{S.30}$$

where $|| \cdot ||_F$ is the Frobenious norm. These lead to

$$\lambda_{max} \leq \alpha \frac{\kappa_1 - \kappa_2}{T} M + O(M^{1-q^*}). \tag{S.31}$$

Finally, sandwiching $\lambda_{max}$ by bounds (S.27) and (S.31), we asymptotically obtain

$$\lambda_{max} \sim \alpha \frac{\kappa_1 - \kappa_2}{T} M, \tag{S.32}$$

in the large $M$ limit.

Note that $\kappa_1 > \kappa_2$ holds in our settings. We can easily observe $\hat{q}_t^l > \hat{q}_{st}^l$ from the Cauchy–Schwarz inequality and it leads to $\kappa_1 > \kappa_2$ (strictly speaking, when $\phi(x)$ is a constant function, its equality holds and we have $\hat{q}_t^l = \hat{q}_{st}^l$ and $\kappa_1 = \kappa_2$. However, we do not suppose the constant function as an "activation" function and then $\kappa_1 > \kappa_2$ holds).

**B.1.3 Largest eigenvalue when $M/T = const.$**

This case requires a careful consideration of the $O(M^{1-q^*})$ term in the reversed FIM (S.21). This is because the non-diagonal term of $K_{L,mBN}$ asymptotically decreases to zero in the large $M$ limit and the $O(M^{1-q^*})$ term becomes non-negligible. We found the following theorem without using the gradient independence assumption;

**Theorem B.1.** *Suppose a non-centered network with the mean subtraction in the last layer (Eq. (15)) and i.i.d. input samples generated by Eq. (3). When $T = O(M)$ with a constant $\rho := M/T$, the largest eigenvalue in the large $M$ limit is asymptotically evaluated as*

$$\rho\alpha(\kappa_1 - \kappa_2) + c_1 M^{1-2q^*} \leq \lambda_{max} \leq \sqrt{C\alpha^2\rho(\kappa_1 - \kappa_2)^2 M + c_2 M^{2(1-q^*)}} \tag{S.33}$$

*for $q^* = \min\{q, 1/2\}$, non-negative constants $c_1$ and $c_2$.*

*Proof.* To evaluate the largest eigenvalue, we use the second moment of the eigenvalues, i.e., $s_\lambda := \sum_i^P \lambda_i^2 / P$. Because $F^*_{L,mBN}$ is positive semi-definite, we have $\sum_i \lambda_i^2 = \sum_{st}((F^*_{L,mBN})_{st})^2$ and obtain

$$s_\lambda = \sum_{k,k'} \sum_{s,t} \left( \alpha \frac{M}{T}(K_{L,mBN})_{st}\delta_{kk'} + \frac{1}{T}(F_0(k,k'))_{st} \right)^2 / P, \qquad (S.34)$$

where we denote the second term of Eq. (S.21) as $F_0 = O(M^{1-q^*})$. We then have

$$s_\lambda =$$

$$\sum_{s,t} \left\{ C\left( \alpha \frac{M}{T}(K_{L,mBN})_{st} \right)^2 + 2\alpha \frac{M}{T^2}(K_{L,mBN})_{st}\sum_k (F_0(k,k))_{st} + \sum_{k,k'} \frac{1}{T^2}(F_0(k,k'))_{st}^2 \right\} / P$$

$$= \frac{C\alpha}{T}(1-1/T)(\kappa_1-\kappa_2)^2 + \frac{2(\kappa_1-\kappa_2)}{MT^2}(1-1/T)\sum_t (K_{L,mBN})_{tt}\sum_k (F_0(k,k))_{tt}$$

$$- \frac{2(\kappa_1-\kappa_2)}{MT^3}\sum_{s\neq t}(K_{L,mBN})_{st}\sum_k (F_0(k,k))_{st} + \sum_{s,t}\sum_{k,k'} \frac{1}{\alpha T^2 M^2}(F_0(k,k'))_{st}^2. \qquad (S.35)$$

When $T = O(M^p)$ ($p \geq 0$), the first term is of $O(1/M^p)$, the second and third terms are of $O(1/M^{p+q^*})$, and the fourth term is of $O(1/M^{2q^*})$. Therefore, the second and third terms are negligible compared to the first term for all $p$ and $q^*$. The fifth term is non-negative by definition. Although we can make the bounds of $\lambda_{max}$ for all $p$, we focus on $p = 1$ for simplicity. In the large $M$ limit, we have asymptotically

$$s_\lambda \sim \frac{C\alpha\rho}{M}(\kappa_1-\kappa_2)^2 + \frac{c_0}{M^{2q^*}}. \qquad (S.36)$$

The constant $c_0$ comes from the fourth term of Eq. (S.35) and is non-negative.

The lower bound of $\lambda_{max}$ is derived from $\lambda_{max} \geq \sum_i \lambda_i^2 / \sum_i \lambda_i = s_\lambda / m_\lambda$, that is,

$$\lambda_{max} \geq (\kappa_1-\kappa_2)\rho + c_1 M^{1-2q^*}. \qquad (S.37)$$

The upper bound comes from $\lambda_{max} \leq \sqrt{\sum_i \lambda_i^2} = \sqrt{Ps_\lambda}$ and we have

$$\lambda_{max} \leq \sqrt{C\alpha^2\rho(\kappa_1-\kappa_2)^2 M + c_2 M^{2(1-q^*)}}. \qquad (S.38)$$

The non-negative constants $c_1$ and $c_2$ come from $c_0$. $\qquad \square$

Thus, we find that $\lambda_{max}$ is of order $M^{1-2q^*}$ at least and of order $M^{1-q^*}$ at most. Since we have $0 < q^* \leq 1/2$ by definition, the order of $\lambda_{max}$ is always lower than order of $M$. Therefore, we can conclude that the mean subtraction alleviates the pathological sharpness for any $q$.

Furthermore, we confirmed that $q = q^* = 1/2$ held in simulations as shown in Fig. S.1. We numerically computed $\tilde{q}^1_{M,st}$ in DNNs with input samples generated by Eq. (3), random Gaussian weights and biases. We set $\alpha_l = C = 1$, $L = 3$ and $T = 100$. The experiments involved 100 different ensembles. We observed that the standard deviation of $\tilde{q}^1_{M,st}$ decreased with the order of $1/\sqrt{M}$, which indicated that $q = 1/2$ held in the real models.

From the theoretical perspective, we found that the gradient independence assumption achieves $q = q^* = 1/2$ and leads to a constant lower bound independent of $M$.

**Lemma B.2.** *The gradient independent assumption yields $q = q^* = 1/2$.*

Figure S.1: Standard deviation of $\tilde{q}^1_{M,st}$. Variances of parameters were given by $(2,0)$ in ReLU case and $(\sigma_w^2, \sigma_b^2) = (3, 0.64)$ in tanh case.

*Proof.* Under the gradient assumption, we can apply the central limit theorem to $\tilde{q}^l_{M,st}$ because we can assume that they do not share the same random weights and biases. That is,

$$\tilde{q}^l_{M,st} = \sum_i \phi_i'^l(s)\phi_i'^l(t) \sum_{j,j'} W_{ji}^{l+1} W_{j'i}^{l+1} \delta_j^{l+1}(s) \delta_{j'}^{l+1}(t) \tag{S.39}$$

$$= \sum_i \phi_i'^l(s)\phi_i'^l(t) \sum_{j,j'} \tilde{W}_{ji}^{l+1} \tilde{W}_{j'i}^{l+1} \delta_j^{l+1}(s) \delta_{j'}^{l+1}(t) \quad \text{(by Assumption 3.2)} \tag{S.40}$$

$$= \sum_i \phi_i'^l(s)\phi_i'^l(t) \frac{\sigma_w^2}{M_l} \sum_j \delta_j^{l+1}(s)\delta_j^{l+1}(t) + O(1/\sqrt{M}) \quad \text{(in the large $M$ limit)} \tag{S.41}$$

$$= \sigma_w^2 \tilde{q}_{st}^{l+1} I_{\phi'}[q_t^l, q_{st}^l] + O(1/\sqrt{M}). \tag{S.42}$$

Thus, we have $\tilde{q}^l_{M,st} = \tilde{q}^l_{st} + O(1/\sqrt{M})$ and obtain $q = q^* = 1/2$. $\qquad\square$

The bounds for $\lambda_{max}$ in Theorem 3.3 are immediately obtained from Theorem B.1 and Lemma B.2.

## B.2 Mean subtraction and variance normalization

Define $\bar{u}_k(t) =: u_k^L(t) - \mu_k(\theta)$. The derivatives of output units are given by

$$\nabla_\theta f_k = \frac{1}{\sigma_k(\theta)} \nabla_\theta \bar{u}_k - \frac{1}{\sigma_k(\theta)^3} \bar{u}_k E[\bar{u}_k \nabla_\theta \bar{u}_k]. \tag{S.43}$$

Then, the FIM is given by

$$F_{L,BN} := \sum_k^C E[\nabla_\theta f_k(t) \nabla_\theta f_k(t)^\top] \tag{S.44}$$

$$= \sum_k^C \frac{1}{\sigma_k(\theta)^2} \left( E[\nabla_\theta \bar{u}_k \nabla_\theta \bar{u}_k^\top] - \frac{1}{\sigma_k(\theta)^2} E[\bar{u}_k \nabla_\theta \bar{u}_k] E[\bar{u}_k \nabla_\theta \bar{u}_k]^\top \right), \tag{S.45}$$

using the fact $\sigma_k(\theta)^2 = E[\bar{u}_k^2]$. We can represent $F_{L,BN}$ in a matrix representation as

$$F_{L,BN} = (R - \bar{R})Q(R - \bar{R})^\top. \tag{S.46}$$

$Q$ is a $CT \times CT$ matrix whose $(k, k')$-th block is given by a $T \times T$ matrix,

$$Q(k, k') := \frac{1}{\sigma_k^2} \left( I_T - \frac{1}{T\sigma_k^2} \bar{u}_k \bar{u}_k^\top \right) \delta_{kk'}, \tag{S.47}$$

where $I_T$ is a $T \times T$ identity matrix, $\sigma_k^2$ means $\sigma_k(\theta)^2$, and $Q(k,k)$ is a projector to the vector $\bar{u}_k$. $F_{L,BN}$ and the following matrix have the same non-zero eigenvalues,

$$F_{L,BN}^* := Q(R - \bar{R})^\top (R - \bar{R}) = QF_{L,mBN}^*. \tag{S.48}$$

$F_{L,BN}^*$ is a $CT \times CT$ matrix and partitioned into $C^2$ block matrices. Using Eq. (S.21), we obtain the $(k, k')$-th block as

$$F_{L,BN}^*(k, k') = \alpha \frac{M}{T} Q(k, k') K_{L,mBN} \delta_{kk'} + \frac{1}{T} O(M^{1-q^*}), \tag{S.49}$$

where the independence assumption yields $q^* = 1/2$. The first term is easy to evaluate,

$$Q(k, k) K_{L,mBN} = \frac{1}{\sigma_k^2} (\kappa_1 - \kappa_2) \left( I_T - \frac{1}{T\sigma_k^2} (1 - \frac{1}{T}) \bar{u}_k \bar{u}_k^\top \right), \tag{S.50}$$

by using the fact of $\sum_t \bar{u}_k^L(t) = 0$. Suppose the case of $\rho = M/T = const$. Regarding the diagonal entries of $Q(k, k') K_{L,mBN}$, the contribution of $\frac{1}{\sigma_k^2 T}(1 - \frac{1}{T}) u_k u_k^\top$ is negligible to that of $I_T$ in the large $T$ limit. Thus, we asymptotically obtain

$$m_\lambda \sim \sum_k \frac{1}{\sigma_k^2} (\kappa_1 - \kappa_2)/M. \tag{S.51}$$

The bounds of the largest eigenvalue are straightforwardly obtained from the second moment as in the deviation of Theorem B.1. Since the second moment $s_\lambda = \sum_i \lambda_i^2 / P$ is given by a trace of the squared matrix in general, we have

$$s_\lambda = \text{Trace}(F_{L,BN}^*{}^2)/P \tag{S.52}$$

$$= \sum_{k,k'} \text{Trace}(F_{L,BN}^*(k, k') F_{L,BN}^*(k', k))/P \tag{S.53}$$

$$= \alpha^2 \rho^2 \sum_k \text{Trace}(Q(k, k) K_{L,mBN} Q(k, k) K_{L,mBN})/P + O(1/M) \tag{S.54}$$

$$= \alpha \rho \sum_k \frac{1}{\sigma_k^4} (\kappa_1 - \kappa_2)^2 / M + O(1/M). \tag{S.55}$$

The lower bound is given by $\lambda_{max} \geq s_\lambda / m_\lambda$ and the upper bound by $\lambda_{max} \leq \sqrt{P s_\lambda}$.

## C  Batch normalization in middle layers

Batch normalization makes the chain of backward signals more complicated as follows. Suppose the $t$-th input sample is given. Then, the activation in each layer depends not only on the $t$-th sample but also on the whole of all samples. This is because batch normalization includes $\mu^l$ and $\sigma^l$, which depend on the whole of all samples in the batch. Therefore, we should compute derivatives as

$$\frac{\partial u_k^L(t)}{\partial W_{ij}^l} = \sum_a \delta_{k,i}^l(t; a) h_j^{l-1}(a), \tag{S.56}$$

where we defined

$$\delta_{k,i}^l(t; a) := \frac{\partial u_k^L(t)}{\partial u_i^l(a)}. \tag{S.57}$$

Its chain rule is given by

$$\delta_{k,i}^l(t; a) = \sum_{b,j} \frac{\partial u_j^{l+1}(b)}{\partial u_i^l(a)} \delta_{k,j}^{l+1}(t; b) \tag{S.58}$$

$$= \frac{1}{\sigma_i^l} \sum_b \phi_i'^l(b) P_i^l(a, b) \sum_j W_{ji}^{l+1} \delta_{k,j}^{l+1}(t; b), \tag{S.59}$$

where we defined

$$P_i^l(a, b) := \delta_{ab} - \frac{1}{T} - \frac{(u_i^l(b) - \mu_i^l)(u_i^l(a) - \mu_i^l)}{(\sigma_i^l)^2 T}. \tag{S.60}$$

Recently, Yang et al. [14] investigated a gradient explosion of the above chain rule in extremely deep networks although it requires a complicated formulation of mean field equations and is analytically intractable in general cases. In the following, we demonstrate an approach to batch normalization in the middle layers by avoiding the complicated analysis of the chain rule.

## C.1 Effect of un-normalized last layer on the FIM

The derivative with respect to the $L$-th layer is independent of the complicated chain of batch normalization because we do not normalize the last layer and have

$$\frac{\partial f_k(t)}{\partial W_{ij}^L} = \sum_a \delta_{k,i}^L(t;a) h_j^{L-1}(a) = h_j^{L-1}(t)\delta_{ki}, \tag{S.61}$$

where we used $\delta_{k,i}^L(t;a) = \delta_{ki}\delta_{ta}$. The lower bound of $\lambda_{max}$ is derived as follows:

$$\lambda_{max} = \max_{||x||^2=1; x\in\mathbb{R}^P} x^\top F x \tag{S.62}$$

$$\geq \max_{||x||^2=1; x\in\mathbb{R}^{CM_{L-1}}} x^\top F_L x \tag{S.63}$$

$$=: \lambda_{max}^L, \tag{S.64}$$

where we denote a diagonal block of $F$ as $F_L := \sum_k \mathrm{E}[\nabla_{\theta^L} f_k \nabla_{\theta^L} f_k^\top]$ and $\theta^L$ is a vector composed of all entries of $W^L$. We denote its largest eigenvalue as $\lambda_{max}^L$. One can represent $F_L = RR^\top$ by using $R := [\nabla_{\theta^L} f_1 \ \nabla_{\theta^L} f_2 \ \cdots \ \nabla_{\theta^L} f_C]/\sqrt{T}$. Its reversed FIM is given by $F_L^* := R^\top R$. In the large $M$ limit, we have

$$F_L^*(k,k') = \alpha\frac{M}{T}K_L\delta_{kk'} + \frac{1}{T}O(M^{1-q^*}). \tag{S.65}$$

The matrix $K_L$ is defined by $(K_L)_{st} := \hat{q}_{t,BN}^{L-1} \ (s=t), \ \hat{q}_{st,BN}^{L-1} \ (s\neq t)$ where we denote feedroward order parameters for batch normalization as

$$\hat{q}_{t,BN}^l := \frac{\sum_i}{M_l}\phi(\bar{u}_i^l(t))^2, \ \hat{q}_{st,BN}^l := \frac{\sum_i}{M_l}\phi(\bar{u}_i^l(t))\phi(\bar{u}_i^l(s)). \tag{S.66}$$

We then have

$$\lambda_{max}^L \geq \nu_k^\top F_L \nu_k = \frac{T-1}{T}\hat{q}_{st,BN}^{L-1} + \frac{\hat{q}_{t,BN}^{L-1}}{T}. \tag{S.67}$$

The evaluation of the order parameters are shown in the following subsection. When the activation function is non-negative, the order paramters are positive. In particular, they are analytically tractable in ReLU networks.

## C.2 Specific values of $\hat{q}_{t,BN}^{L-1}$ and $\hat{q}_{st,BN}^{L-1}$

Order parameters for batch normalization in the middle layers (S.66) require a careful integral over a $T$-dimensional Gaussian distribution [14]. This is because the pre-activation $\bar{u}_i^l$ depends on all of $u_i^l(t)$ $(t=1,...,T)$ which share the same weight $W_{ij}^l$. Therefore, we generally need the integration of $\phi(\bar{u}_i^l(t))$ over the $T$-dimensional Gaussian distribution, that is,

$$\hat{q}_{t,BN}^l = \int Du^l \phi\left(\frac{u^l(t) - \sum_{t'} u^l(t')/T}{\sqrt{\sum_t(u^l(t) - \sum_{t'} u^l(t')/T)^2/T}}\right)^2, \tag{S.68}$$

$$\hat{q}_{st,BN}^l = \int Du^l \phi\left(\frac{u^l(t) - \sum_{t'} u^l(t')/T}{\sqrt{\sum_t(u^l(t) - \sum_{t'} u^l(t')/T)^2/T}}\right)\phi\left(\frac{u^l(s) - \sum_{t'} u^l(t')/T}{\sqrt{\sum_t(u^l(t) - \sum_{t'} u^l(t')/T)^2/T}}\right), \tag{S.69}$$

where $u^l = (u^l(1), u^l(2), ..., u^l(T))$ is a $T$ dimensional vector and $u^l \sim \mathcal{N}(0, \sigma_w^2 \Sigma_{l-1})$. The $T \times T$ covariance matrix is defined by $(\Sigma_{l-1})_{st} = \hat{q}_{st,BN}^{l-1} \ (s \neq t), \ \hat{q}_{t,BN}^{l-1} \ (s = t)$. These order parameters are positive when the activation function is non-negative (strictly speaking, non-negative and $\phi(x) > 0$ for certain $x$).

Although the above integral is analytically intractable in many activation functions, Yang et al. [14] gave profound insight into the integral. For instance, Corollary F.10 in [14] revealed that the ReLU activation is more tractable, and we have

$$\hat{q}_t^l = 1/2, \ \hat{q}_{st}^l = \frac{1}{2}J(-1/(T-1)), \tag{S.70}$$

where $J(x) := (\sqrt{1-x^2} + (\pi - \arccos(x))x)/\pi$ is known as the arccosine kernel. Wei et al. [32] proposed a mean field approximation on the computation of order parameters for batch normalization, which is consistent with the above order parameters in the large $T$ limit. The previous study [14] also proposed some methods to evaluate the order parameters in more general activation functions.

# D  Additional experiment on gradient descent training

Figure S.2: Exhaustively searched training losses depending on $M$ (width) and $\eta$ (learning rate). We trained DNNs over ten trials with different random seeds, and plotted the minimum value of the training loss over all trials. Gray area corresponds to the explosion of the minimum value (i.e., larger than $10^3$) and means that gradient dynamics in all trials exploded in that area. The other experimental settings are the same as those in Fig.2. The theoretical line predicted well the experimental results of wide networks. Mean subtraction achieved the larger area of low training losses.

# E  Layer normalization

## E.1  Order parameters in layer normalization

We show that the order parameters under layer normalization are quite similar to those without the normalization. This is because the random weights and biases make the contribution of layer normalization relatively easy. In the large $M$ limit, we asymptotically have

$$\mu^l(t) = \sum_j \left( \frac{\sum_i W_{ij}^l}{M_l} \right) h_j^{l-1}(t) + \frac{\sum_i b_i^l}{M_l} = 0, \tag{S.71}$$

and

$$\sigma^l(t)^2 = \sum_{j,j'} \frac{\sum_i W_{ij}^l W_{ij'}^l}{M_l} h_j^{l-1}(t) h_{j'}^{l-1}(t) + \frac{\sum_i (b_i^l)^2}{M_l} = \sigma_w^2 \hat{q}_t^{l-1} + \sigma_b^2 \tag{S.72}$$

for $l = 1, ..., L - 1$. Let us denote feedforward order parameters as

$$\hat{q}_t^l := \frac{\sum_i}{M_l} \phi(\bar{u}_i^l(t))^2, \quad \hat{q}_{st}^l := \frac{\sum_i}{M_l} \phi(\bar{u}_i^l(t)) \phi(\bar{u}_i^l(s)). \tag{S.73}$$

The same calculation as in the feedforward propagation without normalization leads to

$$\hat{q}_t^{l+1} = \int Du \phi^2(u), \quad \hat{q}_{st}^{l+1} = I_\phi \left[ 1, \frac{\sigma_w^2 \hat{q}_{st}^l + \sigma_b^2}{\sigma_w^2 \hat{q}_t^l + \sigma_b^2} \right]. \tag{S.74}$$

The backward order parameters are also very similar to those without layer normalization. Let us consider the chain rule which appears in a FIM:

$$\frac{\partial u_k^L(t)}{\partial W_{ij}^l} = \delta_{k,i}^l(t) h_j^{l-1}(t). \tag{S.75}$$

Ommiting index $k$ in $\delta_{k,i}^l(t)$ to avoid complicated notation, we have

$$\delta_i^l(t) = \sum_j \frac{\partial u_j^{l+1}}{\partial u_i^l}(t) \delta_j^{l+1}(t) \tag{S.76}$$

$$= \frac{1}{\sigma^l(t)} \sum_k \phi_k^{\prime l}(t) P_{ki}^l(t) \sum_j W_{jk}^{l+1} \delta_j^{l+1}(t), \tag{S.77}$$

where we define $P^l_{ki}(t) := \frac{\partial \bar{u}^l_k}{\partial u^l_i}(t)$, which is an essential effect of layer normalization on the chain, and it becomes

$$P^l_{ki}(t) = \delta_{ki} - \frac{1}{M_l} - \frac{n^l_k(t)n^l_i(t)}{M_l}, \tag{S.78}$$

where we define $n^l_k(t) := (u^l_k(t) - \mu^l(t))/\sigma^l(t)$. Let us denote backward order parameters as $\hat{q}^l_t := \sum_i \delta^l_i(t)^2$ and $\tilde{q}^l_{st} := \sum_i \delta^l_i(s)\delta^l_i(t)$ in the large $M$ limit. We then have

$$\hat{q}^l_{st} = \frac{1}{\sigma^l(t)\sigma^l(s)} \sum_i \sum_{k,k'} \phi'^l_k(s)\phi'^l_{k'}(t)P^l_{ki}(s)P^l_{k'i}(t) \sum_{j,j'} W^{l+1}_{jk}W^{l+1}_{j'k'}\delta^{l+1}_j(s)\delta^{l+1}_{j'}(t) \tag{S.79}$$

$$= \frac{1}{\sigma^2_w \hat{q}^{l-1}_t + \sigma^2_b} \sum_{k,k'} \phi'^l_k(s)\phi'^l_{k'}(t)\Gamma^l_{k,k'}(s,t) \sum_{j,j'} W^{l+1}_{jk}W^{l+1}_{j'k'}\delta^{l+1}_j(s)\delta^{l+1}_{j'}(t), \tag{S.80}$$

where we substituted $\sigma^l(t) = \sigma^l(s) = \sqrt{\sigma^2_w \hat{q}^{l-1}_t + \sigma^2_b}$ and defined

$$\Gamma^l_{k,k'}(s,t) := \delta_{kk'} - \frac{n^l_k(t)n^l_{k'}(t) + n^l_k(s)n^l_{k'}(s) + 1}{M_l} - \frac{n^l_k(s)n^l_{k'}(t) \sum_i n^l_i(s)n^l_i(t)}{M^2_l}. \tag{S.81}$$

Under the gradient independence assumption, we can replace $W^{l+1}_{jk}$ and $W^{l+1}_{j'k'}$ in Eq. (S.80) with $\tilde{W}^{l+1}_{jk}$ and $\tilde{W}^{l+1}_{j'k'}$ which are freshly generated from $\mathcal{N}(0, \sigma^2_w/M_l)$. This is a usual trick in mean field theory of DNNs [11–14]. In the large $M$ limit, we have

$$\tilde{q}^l_{st} = \frac{1}{\sigma^2_w \hat{q}^{l-1}_t + \sigma^2_b} \sum_k \phi'^l_k(s)\phi'^l_k(t)\Gamma^l_{k,k}(s,t)\frac{\sigma^2_w}{M_l}\tilde{q}^{l+1}_{st}, \tag{S.82}$$

$$\tag{S.83}$$

where

$$\Gamma^l_{k,k}(s,t) = 1 - \frac{n^l_k(t)^2 + n^l_k(s)^2 + 1}{M_l} - \frac{n^l_k(s)n^l_k(t) \sum_i n^l_i(s)n^l_i(t)}{M^2_l}. \tag{S.84}$$

The first term of $\Gamma^l_{k,k}(s,t)$ is dominant in the large $M$ limit because other terms are of order $1/M$. Then, we have

$$\tilde{q}^l_{st} = \frac{1}{\sigma^2_w \hat{q}^{l-1}_t + \sigma^2_b} \sum_k \phi'^l_k(s)\phi'^l_k(t)\frac{\sigma^2_w}{M_l}\tilde{q}^{l+1}_t. \tag{S.85}$$

$$\tag{S.86}$$

After applying the central limit theorem to $\sum_k \frac{\phi'^l_k(s)\phi'^l_k(t)}{M_l}$, we have

$$\tilde{q}^l_t = \frac{\sigma^2_w \tilde{q}^{l+1}_t}{\sigma^2_w \hat{q}^{l-1}_t + \sigma^2_b} \int Du[\phi'(u)]^2, \quad \tilde{q}^l_{st} = \frac{\sigma^2_w \tilde{q}^{l+1}_{st}}{\sigma^2_w \hat{q}^{l-1}_t + \sigma^2_b} I_{\phi'}\left[1, \frac{\sigma^2_w \hat{q}^{l-1}_{st} + \sigma^2_b}{\sigma^2_w \hat{q}^{l-1}_t + \sigma^2_b}\right]. \tag{S.87}$$

## E.2  FIM

### E.2.1  Effect of the normalization in the last layer

Denote the mean subtraction in the last layer as $\bar{u}_k(t) =: u^L_k(t) - \mu^L(t)$. The derivatives in the last layer are given by

$$\nabla_\theta f_k(t) = \frac{1}{\sigma(t)}\nabla_\theta \bar{u}_k(t) - \frac{1}{C\sigma(t)^3}\bar{u}_k(t)\sum_i \bar{u}_i(t)\nabla_\theta \bar{u}_i(t), \tag{S.88}$$

where $\sigma(t)^2 := \sum_k \bar{u}_k(t)^2/C$. Then, the FIM is given by

$$F_{LN} := \sum_k \mathrm{E}[\nabla_\theta f_k(t)\nabla_\theta f_k(t)^\top] \tag{S.89}$$

$$= \mathrm{E}\left[\frac{1}{\sigma(t)^2}\sum_k \nabla_\theta \bar{u}_k(t)\nabla_\theta \bar{u}_k(t)^\top - \frac{1}{C\sigma(t)^4}\sum_{k,k'} \bar{u}_k(t)\bar{u}_{k'}(t)\nabla_\theta \bar{u}_k(t)\nabla_\theta \bar{u}_{k'}(t)^\top\right]. \tag{S.90}$$

We can represent $F_{L,BN}$ in a matrix representation. Define a $P \times CT$ matrix $R$ by

$$R := \frac{1}{\sqrt{T}}[\nabla_\theta u_1^L \ \nabla_\theta u_2^L \ \cdots \ \nabla_\theta u_C^L]. \tag{S.91}$$

Its columns are the gradients on each input sample, i.e., $\nabla_\theta u_k^L(t)$ $(t = 1, ..., T)$. We then have

$$F_{LN} = (R - \bar{R})Q(R - \bar{R})^\top, \tag{S.92}$$

where $\bar{R}$ is defined as a $CT \times P$ matrix whose $((k-1)T+t)$-th column is given by a vector $\nabla_\theta \mu^L(t)$ $(t = 1, ..., T, k = 1, ..., C)$. We also defined a $CT \times CT$ matrix $Q$ whose $(k, k')$-th block matrix is given by the following $T \times T$ matrix:

$$Q(k, k')_{st} := \frac{1}{\sigma(t)^2}\left(\delta_{kk'} - \frac{1}{C\sigma(t)^2}\bar{u}_k^L(t)\bar{u}_{k'}^L(t)\right)\delta_{st} \tag{S.93}$$

for $k, k' = 1, ..., C$. This $Q(k, k')$ is a diagonal matrix. Compared to the matrix $Q$ in batch normalization (Eq. (S.47)), $Q$ in layer normalization is not block-diagonal. This is because layer normalization in the last layer yields interaction between different output units.

We introduce the following matrix which has the same non-zero eigenvalues as $F_{LN}$:

$$F_{LN}^* = Q(R - \bar{R})^\top(R - \bar{R}) = QF_{mLN}^*, \tag{S.94}$$

where

$$F_{mLN}^* := (R - \bar{R})^\top(R - \bar{R}). \tag{S.95}$$

This $F_{mLN}^*$ corresponds to the mean subtraction in layer normalization. Its entries are given by

$$F_{mLN}^*(k, k')_{st} := (\nabla_\theta u_k^L(s) - \nabla_\theta \mu^L(s))^\top(\nabla_\theta u_{k'}^L(t) - \nabla_\theta \mu^L(t)) \tag{S.96}$$

$$= u_k^L(s)^\top\nabla_\theta u_{k'}^L(t) - \frac{\sum_a}{C}\nabla_\theta u_k^L(s)^\top\nabla_\theta u_a^L(t) - \frac{\sum_a}{C}\nabla_\theta u_a^L(s)^\top\nabla_\theta u_{k'}^L(t)$$

$$+ \frac{\sum_a}{C^2}\nabla_\theta u_a^L(s)^\top\nabla_\theta u_a^L(t) + \frac{\sum_{a \neq a'}}{C^2}\nabla_\theta u_a^L(s)^\top\nabla_\theta u_{a'}^L(t). \tag{S.97}$$

In the large $M$ limit, we have

$$\nabla_\theta u_k^L(s)^\top\nabla_\theta u_{k'}^L(t) = \sum_{l,ij}(\delta_{k,i}^l(s)\delta_{k',i}^l(t))(h_j^{l-1}(s)h_j^{l-1}(t)) \tag{S.98}$$

$$= \sum_l M_l\tilde{q}_{st}^l\hat{q}_{st}^{l-1}\delta_{kk'} + O(M^{1-q^*}), \tag{S.99}$$

after doing the same calculation as Eq. (S.12) and using the order parameters obtained in Section E.1. We have $q^* = 1/2$ due to the gradient independence assumption. The reversed FIM becomes

$$F_{mLN}^*(k, k') = \left(\delta_{kk'} - \frac{1}{C}\right)\alpha\frac{M}{T}K_{LN} + \frac{1}{T}O(M^{1-q^*}), \tag{S.100}$$

where we defined a matrix $K_{LN}$ by

$$(K_{LN})_{st} = \kappa_1' \ (s = t), \quad \kappa_2' \ (s \neq t), \tag{S.101}$$

with

$$\kappa_1' := \sum_{l=1}^L \frac{\alpha_{l-1}}{\alpha}\tilde{q}_t^l\hat{q}_t^{l-1}, \quad \kappa_2' := \sum_{l=1}^L \frac{\alpha_{l-1}}{\alpha}\tilde{q}_{st}^l\hat{q}_{st}^{l-1}. \tag{S.102}$$

Note that the order parameters $(\hat{q}_t^l, \hat{q}_{st}^l, \tilde{q}_t^l, \tilde{q}_{st}^l)$ of layer normalization are computed by the recurrence relations (Eqs. (S.74) and (S.87)). Finally, the $(k, k')$-th block of $F_{LN}^*$ is given by

$$F_{LN}^*(k, k') = \sum_a Q(k, a)F_{mLN}^*(a, k') \tag{S.103}$$

$$= \alpha\frac{M}{T}\text{diag}\left(\frac{1}{\sigma(t)^2}\right)\left(\left(\delta_{kk'} - \frac{1}{C}\right)I_T - \frac{1}{C}\text{diag}\left(\frac{\bar{u}_k(t)\bar{u}_{k'}(t)}{\sigma(t)^2}\right)\right)K_{LN}, \tag{S.104}$$

where $\text{diag}(f(t))$ means a $T \times T$ diagonal matrix whose $t$-th diagonal entry is $f(t)$.

### E.2.2 Eigenvalue statistics

The mean is asymptotically given by

$$m_\lambda = \text{Trace}(F^*_{LN})/P \tag{S.105}$$

$$= \sum_k \text{Trace}(F^*_{LN}(k,k))/P \tag{S.106}$$

$$\sim \eta_1 \frac{(C-2)\kappa'_1}{M}, \tag{S.107}$$

where $\eta_1 := \frac{1}{T}\sum_t \frac{1}{\sigma(t)^2}$.

The largest eigenvalue is evaluated using the second moment of the eigenvalues. Since the second moment $s_\lambda = \sum_i \lambda_i^2/P$ is given by a trace of the squared matrix in general, we have

$$s_\lambda = \text{Trace}((F^*_{LN})^2)/P \tag{S.108}$$

$$= \sum_k \text{Trace}(\sum_a F^*_{LN}(k,a)F^*_{LN}(a,k))/P. \tag{S.109}$$

We obtain

$$((\sum_a F^*_{LN}(k,a)F^*_{LN}(a,k))_{tt} \tag{S.110}$$

$$= (\alpha\frac{M}{T})^2 \sum_{a,t'} \frac{1}{\sigma(t)^2}\left(\left(\delta_{ka} - \frac{1}{C}\right) - \frac{1}{C}\left(\frac{\bar{u}_k(t)\bar{u}_a(t)}{\sigma(t)^2}\right)\right)(K_{LN})_{tt'} \tag{S.111}$$

$$\cdot \frac{1}{\sigma(t')^2}\left(\left(\delta_{ak} - \frac{1}{C}\right) - \frac{1}{C}\left(\frac{\bar{u}_a(t')\bar{u}_k(t')}{\sigma(t')^2}\right)\right)(K_{LN})_{t't}$$

$$= \sum_{t'}(\alpha\frac{M}{T})^2 \frac{1}{\sigma(t)^2\sigma(t')^2}(C-3+g(t,t')^2)((\kappa_1'^2 - \kappa_2'^2)\delta_{tt'} + \kappa_2'^2), \tag{S.112}$$

where we define $g(t,t') := \frac{1}{C}\frac{\sum_a \bar{u}_a(t)\bar{u}_a(t')}{\sigma(t)\sigma(t')}$. Substituting Eq. (S.112) into Eq. (S.109), we obtain

$$s_\lambda = \alpha\frac{(\eta_3^2 - \eta_1^2)T + (C-2)(\eta_1^2 T - \eta_2)}{T}\kappa_2'^2 + \alpha(C-2)\eta_2\frac{\kappa_1'^2}{T}, \tag{S.113}$$

where $\eta_2 := \frac{\sum_t}{T}\frac{1}{\sigma(t)^4}$ and $\eta_3 := \frac{1}{T^2}\sum_{t,t'}\frac{g(t,t')}{\sigma(t)\sigma(t')}$. The lower bound is given by $\lambda_{max} \geq s_\lambda/m_\lambda$ and the upper bound by $\lambda_{max} \leq \sqrt{Ps_\lambda}$.

**Remark on $C = 2$:** Because we have a special symmetry, i.e., $\bar{u}_1(t) = -\bar{u}_2(t) = (u_1^L(t) - u_2^L(t))/2$ in $C = 2$, the gradient (Eq. (S.88)) becomes zero. This is caused by the mean subtraction and variance normalization in the last layer. This makes the FIM a zero matrix. The case of $C > 3$ is non-trivial and the FIM becomes non-zero, as we revealed. Similarly, the gradient (Eq. (S.43)) in batch normalization becomes zero when $T = 2$ due to the same symmetry [14]. Such an exceptional case of batch normalization is not our interest because we focus on the sufficiently large $T$ in Eq. (19).