[Reviews · NeurIPS 2019]

Reviewer 1



This well-written paper is the latest in a series of works which analyze how signals propagate in random neural networks, by analyzing mean and variance of activations and gradients given random inputs and weights. The technical accomplishment can be considered incremental with respect to this series of works. However, while the techniques used are not new, the performed analysis leads to new insights on the use of batch/layer normalization. In particular, the analysis provides a close look on mechanisms that lead to pathological sharpness on DNNs, showing that the mean subtraction is the main ingredient to counter these mechanisms. While these claims would have to be verified in more complicated settings (e.g. with more complicated distributions on inputs and weights), it is an important first step to know that they hold for such simple networks. Minor points follow: - on the sum in E(\theta) (end of page 3), upper limit is defined but not lower one - \lambda_max seems to concentrate, is it possible to compute an average value rather than a bound? - worth adding a comment about the difference between (20)/(21) and (23)/(24) - the normalization techniques are studied considering the whole dataset, and not batches as typically done in SGD; is there any way of adapting it? - I wonder if the authors have any comment on Kunster et al., "Limitations of the Empirical Fisher Approximation", who show that the empirical FIM is not necessarily representative of the geometry of the model # Post-rebuttal addendum I was previously unaware of the publication of [21] at AISTATS, and as such I share my concern with other reviewers regarding the originality of this work. I recommend the clarifications given in the response are included in an eventual updated version of this manuscript. Otherwise, I am quite pleased by the effort the authors have put in the rebuttal, and in particular with the experiments on trained networks. I think this is yet another evidence that this work might be useful in practice. Following these two points⁠—one negative and one positive⁠—I have decided to keep my score the same.

Reviewer 2



Originality: In terms of overall technique, this paper uses techniques that have already been developed in previous mean field papers. However, the application of these techniques to the fisher information is a novel, challenging, and worthwhile contribution. Moreover, looking at the fisher information in combination with normalization techniques is a sufficiently novel perspective that it leads me to believe this paper is sufficiently novel to warrant publication. Quality: I have somewhat mixed opinions about this paper in terms of quality (which has overlap with its significance). On the one hand, the authors have carefully considered a number of different approaches to normalization theoretically. Indeed, I think the distinction between the different normalization settings applied to the spectrum of the fisher is the key takeaway message of the paper. Having said that, the experimental portion of the paper could use work. In particular, the authors remark early on that one should expect the maximum achievable learning rate to scale as the inverse maximum eigenvalue of the fisher. Moreover, batch normalization in the final layer is shown to reduce the maximum eigenvalue. It would be nice to combine these results to show that this improved conditioning does indeed translate to a higher maximum achievable learning rate. Incidentally, it would also be nice to confirm the max eigenvalue behavior of the various normalization types. Clarity: I thought the clarity of the exposition was good. I was able to follow the results and understand the important aspects of the paper clearly. I also feel that given the exposition, if I were to sit down I could probably reproduce most of the results in a reasonable amount of time. One minor point, as I mentioned above, I think an important technique that is introduced in the paper is the duality between the fisher and the NTK (F vs F*). I might recommend the authors move a discussion of this into the main text. Significance: I think the results _could_ be significant. In particular, I think the notion of normalizing the last layer is quite novel (especially since one need only perform mean subtraction). I also think that the theoretical techniques introduced here may end up being impactful. As elaborated above, however, I think to maximize impact the authors should add significant experimental work either to this paper or in a followup work. --------- Update: After reading the author's rebuttal and a subsequent discussion I have decided to raise my review to a 7. I think there is sufficient novelty w.r.t. [21] and the new experiment seems interesting!

Reviewer 3



This paper is well written and properly cited. The interplay between batch normalization and the behavior of Fisher information is interesting. Though, it is questionable whether this calculation explained batch normalization helps optimization. The authors calculated the FIM of training loss at initialization, and it is not the same as the Hessian of training loss. Therefore, it is questionable to say FIM reflects the "landscape" of the training loss function. Also, whether suppressing the extreme eigenvalues of FIM can also make the Hessian (at initialization, and along the whole trajectory) well-conditioned is not clear. Hence, there is a gap in showing batch normalization providing a larger learning rate. Is it due to technical restriction that one cannot calculate the behavior of eigenvalues of Hessian at initialization? In terms of technical contribution, is there some new ingredients in the calculations of batch normalization settings, or it is just a simply re-application of calculations in [1]? [1] Ryo Karakida, Shotaro Akaho, and Shunichi Amari. Universal statistics of fisher information in deep neural networks: Mean field approach. -------- I am satisfied with the authors' response to my questions. I am also glad to see the authors did more experiments in the response. I think I can increase my score to 7. Though, I think the "landscape" terminology used in this paper is not accurate. The FIM is dual to NTK. By the NTK point of view, the dynamics of neural networks is roughly linear, and what matters the convergence is the condition number of the NTK (sharing same eigenvalues as the FIM). The "local shape of landscape" terminology may confuse the readers. I think the authors could explain this more clear in the paper.

[Author Response · NeurIPS 2019]

We thank three reviewers for admitting the importance of our work. We hope that the following explanation will help the reviewers to recognize the novelty and significance of our paper further.

**To Reviewer 1:** The minor points you provided are also of our concern and we will remark them in Discussion. Although the empirical Fisher is not necessarily representative of the geometry (especially, when the number of samples is very small), it always determines NTK's eigenvalues through a dual representation $F^*$. The NTK determines the optimization of sufficiently wide DNNs. We appreciate if you would read our responses to Reviewer 2 and 3. The experiment shown in the below will also respond to your concern on trained networks.

**To Reviewers 2 & 3: Difference of this paper from Karakida, Akaho & Amari, AISTATS2019 [21]**

We would like to emphasize that our work greatly differs from [21]. [21] has **not** shown any result on normalization methods. Moreover, our work is not just a simple re-application of calculation in [21]. Technically speaking, [21] evaluated the first-order term of $F^*$ and neglected lower order term (i.e., the second term in (S.6)). In contrast, our paper enables researchers to evaluate this second term. It is essential because the batch normalization makes the first term comparable to the second term and requires careful evaluation of the second term. In particular, we found that a new quantity, i.e., the convergence rate $q$ (or $q^*$), plays an essential role in the second term and newly developed a framework to evaluate it in Section B.1.3. This enlightened the new direction of theory and enabled us to give the novel insight into the use of normalization methods.

**To Reviewer 2: Experiments on training DNNs and learning rates necessary for convergence**

We agree that experiments will further increase our contribution. As Reviewer 2 recommended, we add an experimental result on the training with the steepest gradient descent argued in Section 5. We did it in the same setting as [21]; we trained DNNs with various widths by using various fixed learning rates, providing i.i.d. Gaussian input samples and labels generated by corresponding teacher networks. Our Fig. (a) is just a reproduction of Fig. 2 (left) in [21]. The theoretical value $\eta = 2/\lambda_{max}$ (Eq.27) computed on the FIM at random initialization predicted well the learning rate necessary for the gradient method to converge. An impressive result is Fig. (b). We confirmed that the batch normalization (mean subtraction) in the last layer allows larger learning rates for convergence and they are independent of width. This result coincides well with Reviewer 2 expectation. Technically speaking, we computed the red line in Fig. (b) by using the lower bound of $\lambda_{max}$, i.e., $\eta = 2/(\rho\alpha(\kappa_1 - \kappa_2))$.

Figure 1: Exhaustively searched training losses depending on $M$(width) and $\eta$ (learning rate). We trained deep ReLU networks for 1000 steps. Losses **exploded in gray area** (i.e., were larger than $10^3$) and red lines show theoretical values of $2/\lambda_{max}$. Experimental setting: $\alpha_l = C = 1$, $L = 3, T = 1000, (\sigma_w^2, \sigma_b^2) = (4, 1)$.

One can also suppose many other experiments related to our theory, but they are too many to enclose in a single paper. Besides, experimental studies in more large-scale networks and datasets are not so easy task because of the computational cost of the huge FIM. So, we expect that our theory and the above experiment will encourage many researchers openly discuss and study the possibility of our results in follow-up works.

**To Reviewer 3: The NTK (neural tangent kernel) determines optimization and loss landscape in wide DNNs**

Answering to your concern, we would like to emphasize recent findings on NTK shown in Jacot et.al., NeurIPS2018 (cited as [19]) and Lee et al. arXiv2019 (cited as [20]). In particular, the work [20] clearly proved that the sufficiently wide DNNs works as a linear model expanded around random initialization $\theta_0$:

$$f(x; \theta_t) = f(x; \theta_0) + \nabla_\theta f(x; \theta_0)^\top \omega_t, \tag{1}$$

where $\omega_t := \theta_t - \theta_0$ and $t$ means the step of the gradient descent shown in lines 284-288. [20] proved a surprising fact that $\omega_t$ is sufficiently small for any $t > 0$ and the network can achieve a zero training error in the large $M$ limit. This means that there is always a global minimum sufficiently close to random initialization. Therefore, the optimization of the wide DNN becomes a convex problem and the loss landscape becomes convex. This convexity is also proved in [19] and the FIM at random initialization determines the loss landscape through a quadratic form $\omega_t^\top F \omega_t$. Thus, its second derivative (Hessian) coincides with the FIM in the large $M$ limit. We hope that the above additional explanations will further clarify our paper's significance, and we will appreciate if you could increase your score.

[Meta-Review · NeurIPS 2019]

The paper is well-written paper and analyzes how signals propagate in random neural networks. It does so by analyzing mean and variance of activations and gradients, given random inputs and weights. The technical contributions are okay, and the analysis leads to new insights on the use of batch/layer normalization.